# Submicron Topographically Patterned 3D Substrates Enhance Directional Axon Outgrowth of Dorsal Root Ganglia Cultured Ex Vivo

**DOI:** 10.3390/biom12081059

**Published:** 2022-07-30

**Authors:** Michele Fornaro, Christopher Dipollina, Darryl Giambalvo, Robert Garcia, Casey Sigerson, Harsh Sharthiya, Claire Liu, Paul F. Nealey, Kolbrun Kristjansdottir, Joshua Z. Gasiorowski

**Affiliations:** 1Department of Anatomy, College of Graduate Studies, Midwestern University, Downers Grove, IL 60515, USA; harshsharthiya@gmail.com; 2Chicago College of Osteopathic Medicine, Midwestern University, Downers Grove, IL 60515, USA; dipollina@gmail.com (C.D.); darrylgiambalvo@gmail.com (D.G.); cdsigerson@gmail.com (C.S.); 3Department of Biomedical Sciences, College of Graduate Studies, Midwestern University, Downers Grove, IL 60515, USA; rgarcia42@midwestern.edu (R.G.); kkrist@midwestern.edu (K.K.); 4Pritzker School of Molecular Engineering, University of Chicago, Chicago, IL 60637, USA; clairel@u.northwestern.edu (C.L.); nealey@uchicago.edu (P.F.N.)

**Keywords:** axonal growth, directionality, biophysical cues, submicron topography, nerve regeneration, sensory fibers

## Abstract

A peripheral nerve injury results in disruption of the fiber that usually protects axons from the surrounding environment. Severed axons from the proximal nerve stump are capable of regenerating, but axons are exposed to a completely new environment. Regeneration recruits cells that produce and deposit key molecules, including growth factor proteins and fibrils in the extracellular matrix (ECM), thus changing the chemical and geometrical environment. The regenerating axons thus surf on a newly remodeled micro-landscape. Strategies to enhance and control axonal regeneration and growth after injury often involve mimicking the extrinsic cues that are found in the natural nerve environment. Indeed, nano- and micropatterned substrates have been generated as tools to guide axons along a defined path. The mechanical cues of the substrate are used as guides to orient growth or change the direction of growth in response to impediments or cell surface topography. However, exactly how axons respond to biophysical information and the dynamics of axonal movement are still poorly understood. Here we use anisotropic, groove-patterned substrate topography to direct and enhance sensory axonal growth of whole mouse dorsal root ganglia (DRG) transplanted ex vivo. Our results show significantly enhanced and directed growth of the DRG sensory fibers on the hemi-3D topographic substrates compared to a 0 nm pitch, flat control surface. By assessing the dynamics of axonal movement in time-lapse microscopy, we found that the enhancement was not due to increases in the speed of axonal growth, but to the efficiency of growth direction, ensuring axons minimize movement in undesired directions. Finally, the directionality of growth was reproduced on topographic patterns fabricated as fully 3D substrates, potentially opening new translational avenues of development incorporating these specific topographic feature sizes in implantable conduits in vivo.

## 1. Introduction

In physiological conditions, sensory and motor fibers of a peripheral nerve are singularly wrapped and separated from each other by the endoneurium, the innermost layer of connective tissue. Within the fiber, the axon is further protected and isolated in the wrapping layers of Schwann cells, which buffer the exposure of the axon to the basal lamina of the endoneurium and its extracellular matrix (ECM). Minimal exposure to the basal lamina occurs at the level of the nodes of Ranvier [1,2,3]. Therefore, each axon is maintained in a tightly controlled microenvironment with little to no exposure to the variety of cells and the chemical and physical milieu these cells are immersed into. An injury to the peripheral nerve disrupts the intimate relation between the axon and the Schwann cell wrapping. Severed axons from the proximal nerve stump are capable of regenerating, but axons are exposed to a completely new environment and will have to regrow surfing on an unexperienced landscape, both chemically and physically. The lack of intimacy with the axon will induce Schwann cells to undergo morphological and functional changes leading to a regressed multipotent stage characterized by active proliferation and migration [4]. While macrophages recruited from blood vessels will actively phagocytize cell debris and clear the area at the site of the lesion, [5] Schwann cells and fibroblasts from the connective tissue will invade the site of the lesion to create a highly organized topographical pattern that favors axonal regrowth [5,6,7]. Recruited cells actively produce and deposit molecules, including growth factor proteins and fibrils in the ECM, thus changing the chemical environment. Moreover, the substrates created by these cells and the rearrangement of the ECM also define new physical properties, with a variety of geometrically defined topographical cues spanning from nano- to micro-scale order. The arrangement and geometries of proteins and bound cues such as netrin, ephrins and semaphorins will influence the axon and its tight interaction with the surface and its chemical and physical properties [8]. The growth cone located at the tip of regenerating axons is essential for the environmental exploration and locomotion of the extending neurite. The cell membrane of the growth cone contains receptors that can sample the molecular milieu, detect a variety of stimuli, explore physical and mechanical cues within the surroundings and direct the axon along a defined path [8,9,10,11]. The mechanical cues of the substrate are used as guides to orient growth or change the direction of growth in response to impediments or cell surface topography. The regenerating neuron senses the environmental cues and transduces mechanical stimuli into gene expression changes and alteration of biochemical signals, thus modifying cell metabolism [12,13,14]. The subsequent response to the cue or stimulus results in axonal outgrowth, possibly over distances of hundreds of microns, to reconnect to the end-tissue “target” [15,16]. Pioneering evidence on how physical cues can modify neuronal cell behavior go back to 1934 and 1947 when Dr. Weiss showed that aligned topographical cues can serve as a track to guide regenerating sensory axons [17].

Limitations in current nerve regeneration techniques have stimulated the development of novel strategies to mimic the extrinsic cues available in the natural nerve environment. In particular, the importance of physical substrate properties such as stiffness, texture and topography and their influence on orienting axonal growth [18,19,20,21]. The ability to manufacture and reproduce nano/microscale topography on implantable materials has opened a new method of enhancing neuritogenesis and peripheral nerve regeneration [22,23,24]. Chemical and physical cues have been adopted to create bioengineered and biomimetic tridimensional scaffolds for nerve repair and axonal growth [25,26,27].

Indeed, nano- and micropatterned substrates have been used as tools to guide axons along a defined path and demonstrate their capacity to respond to topographical features in their microenvironments [28,29,30,31,32]. However, exactly how axons respond to biophysical information and the dynamics of axonal movement are still poorly understood.

Further studies on biomaterials have shown that substrate materials with defined biophysical cues, such as phosphate glass fibers, nano-pillars, aligned monodomain gels and micro-patterned substrates, are able to direct and enhance neuron fiber regeneration [22,33,34]. Guidance-based therapies have made use of nerve grafts, biomimetic conduits enriched with tissue extract and extracellular matrix proteins such as laminin, fibronectin and collagen and synthetic biodegradable materials, or electrically active and conductive materials [35].

In contribution to this promising field aimed at enhancing nerve regeneration by recreating biophysical substrates favorable to the axonal directional growth, this study focuses on the use of groove-patterned substrate topography to direct and enhance sensory axonal growth in a whole mouse dorsal root ganglia (DRG) transplanted ex vivo. Our experiments documented the interaction between regenerating axons of an in toto explant and the geometrical pattern featured as parallel grooves with individual feature sizes between 200–2000 nm. Our results show significantly enhanced and directed growth of the DRG sensory fibers on the 2-dimensional topographic substrates ex vivo compared to a 0 nm pitch (flat) control surface, but only on certain feature sizes. We further assessed the dynamics of axonal movement using time-lapse microscopy. In addition, the study used the best-performing topographic surface in conjunction with a variety of growth factors (BDNF, GDNF, NT3 and NGF) to explore the optimal synergic effects of biophysical and biochemical cues for growth enhancement. Finally, the directionality of growth was reproduced in a 3-D model featuring a similar topographic pattern as per the hemi-3-D planar model, opening new translational avenues of development incorporating these specific topographic feature sizes in implantable conduits *in vivo*.

## 2. Materials and Methods

### 2.1. Fabrication of Topographic Surfaces

Silicon master chips containing repeating ridge and groove patterns were fabricated using X-ray lithography as previously described [36,37,38]. The masters contained ridge and groove areas with pitches of 400, 800, 1400 and 4000 nm (pitch = ridge width + groove width in a 1:1 ridge to groove ratio) and a depth of 300 nm. The silicone masters were used as templates for soft lithography replication to create poly(dimethylsiloxane) (PDMS) stamps with different pitch sizes, along with a flat control stamp, which was effectively a 0 nm pitch. The PDMS stamps were cut into 3 cm × 3 cm squares and stamped onto a thin layer of NOA81 optical adhesive (Norland Products, Cranbury, NJ, USA) that was spin-coated onto 60 mm cell culture dishes. The NOA81 was crosslinked with 365 nm UV light (UVP, Upland, CA, USA) for 100 min and the stamps were carefully peeled off, transferring the 3-dimensional topographic pattern, or 0 nm pitch (flat) control, to the cell culture dish. All cell culture dishes were washed 3 times in sterile phosphate-buffered saline (PBS, pH 7.4, Gibco ThermoFisher, Waltham, MA, USA) and then sterilized by 20 min of 280 nm UV light within a biosafety cabinet prior to use. Surface topography was verified using atomic force microscopy (AFM) on a Bruker Multimode III in tapping mode with a triangular pointed tip (Bruker Corporation, Billerica, MA, USA).

### 2.2. Preparation of Dorsal Root Ganglia (DRG) for Ex Vivo Growth

Adult NIH/Swiss mice of both genders (>50 mice total) were used for this study. All procedures were approved by, and conducted in compliance with, the guidelines of the Institutional Animal Care and Use Committee of Midwestern University. All mice were sacrificed by CO_2_ asphyxiation followed by decapitation. DRG were harvested and prepared as organotypic cultures ex vivo as previously described [39]. Briefly, the pelt was removed from the back and the entirety of the vertebral column was resected from the body. Surgical scissors were inserted into the column and two cuts were made on the dorsolateral sides that run the entire length of the spine, revealing the spinal cord. A surgical microscope was used to aid in the visualization of the DRG located on both sides of the spinal cord. DRG were extricated using forceps and placed in ice-cold F12 media (Gibco Thermofisher, Waltham, MA, USA) until plating. Matrigel (Corning, Inc., Painted Post, NY, USA) was mixed with serum-free media (SFM) in a 1:1 ratio. A total of 10 µL of Matrigel/SFM was pipetted onto the topographical surface per DRG. This dilute Matrigel solution was used to initially adhere the DRG bodies to the plate, and it was largely contained within a 200 µm radius of the DRG body as a thin coating that did not visibly alter the surface as viewed with phase contrast imaging. The plate was covered and placed in the incubation chamber at 37 °C and 5% CO_2_ for 30 to 90 min. Upon matrigel polymerization, the plates were removed from the incubation chamber and returned to the biosafety hood. Four DRGs were plated evenly spaced apart on each pre-coated topographical plate. DRGs were maintained in serum-free media containing 5 ng/mL of nerve growth factor (NGF) (Alomone Labs, Ltd., Jerusalem, Israel) and incubated at 37 °C with 5% CO_2_ for up to 6 days. Media was replaced every 72 h. In experiments where the growth factor was BDNF (Alomone Labs, Ltd.), NT3 (Cell Guidance Systems, Ltd., Cambridge, UK) or GDNF (Sigma-Aldrich, Saint Louis, MO, USA), 5 µg/mL of those factors replaced the NGF in the media.

### 2.3. Immunofluorescence Labeling and Confocal Imaging

On day 6 post-plating, the DRGs were fixed in 4% formalin for 1 h and washed three times in PBS. The entire plate was incubated overnight at 4 °C with primary antibodies, rabbit anti-beta-tubulin (1:2000, Sigma-Aldrich) and mouse anti-peripherin (1:1000, EMD Millipore, Billerica, MA, USA), diluted in PBS with 1% Triton X-100 (Sigma-Aldrich) and 5% normal goat serum (Vector Labs, Newark, CA, USA). The plates were then washed three times in PBS and incubated for 1 h with a secondary antibody solution containing Alexa 488 goat anti-mouse IgG (Life technologies, Carlsbad, CA, USA 1:400) and CY3 goat anti-rabbit IgG (LiStarFish, Milano, Italy 1:400). Explants were then washed three times with PBS and stored at 4 °C bathed in PBS waiting for confocal imaging. An inverted Nikon A1R confocal microscope (Nikon, Melville, NY, USA) was used to capture stitched z-stacks and create complete DRG images with Nikon Elements software (Nikon, Melville, NY, USA, version 5.11), a 10× objective (NA = 0.45) and 488 nm and 561 nm lasers.

### 2.4. Image Analysis of Axonal Growth Parameters

The quantification of axonal growth was performed with ImageJ software (National Institutes of Health, Bethesda, Maryland, USA, version 1.51). First, the body of the DRG was enclosed in a rectangular figure. The dimensions of the enclosing rectangle were based on the intersection of drawing 4 tangent lines on the perimeter of the DRG body at 0, 90, 180 and 270 degrees. The center of the rectangle was used as a point of reference to define distances of 200 µm, 1000 µm and 1500 µm landmarks in the four cardinal directions parallel and perpendicular to the grooved topography surface. A digital grid depicting these landmarks was superimposed over the confocal images, as seen in Appendix A. DRG on 0 nm pitch (flat) control surfaces were assigned arbitrary parallel and perpendicular axes.

ImageJ software was used to quantify all axons crossing each landmark, in each cardinal direction. To be counted, the fluorescence-labeled axons had to be visible on both sides of the landmark and distinguishable from other axons. Indiscernibly bundled axons were counted as one. Axons that curled back and forth across a boundary line multiple times were rarely found, and they were only counted once. Axons that did not cross the 200 µm boundary were not counted for the purposes of these assessments.

The maximal axonal growth parameter was calculated as the average of the 3 longest axons measured with a straight vector line from the center of the rectangle enclosing the DRG to the terminus of the axon. The axonal density was calculated as the number of axons at a given distance marker from the DRG body. The directionality of axonal growth was calculated from the comparison of the number of axons in the north and south quadrants (perpendicular to topography) to the number of axons in the east and west quadrants (parallel to topography) at the 1000 µm and 1500 µm boundary lines.

### 2.5. Live Cell Imaging and Analysis

Between 72–96 h post-plating, DRGs seeded onto 0 nm pitch (flat) or topographically patterned plates were imaged with a 20× objective on an EVOS FL Auto microscope (Manufacturer, City, State abb if USA or Canada, Country,) outfitted with an incubation chamber that maintained an environment of 100% humidity, 5% CO_2_ and 37 °C. Beacon locations to obtain time-lapse images were chosen to ensure there was 20% overlap between fields of view, with at least one image having a part of the DRG body in view. Phase contrast images were obtained every 6 min over the course of 24 h.

The phase contrast images from each beacon were stitched together using the automated portion of the Grid/Collection Stitching plugin in ImageJ, which correlated similarities of bright and dark spots between overlapping images to determine where to place fields of view in x and y [40]. Occasionally, the automated stitching overlap was manually corrected for the time-lapse phase contrast images. The axons were manually tracked, frame by frame, via the MTrackJ ImageJ plugin [41]. The leading edge of the axon growth cone was marked to denote its position, and the next frame was shown to repeat the position tracking over time until the last frame was reached, or until their positions became unclear through an occlusion or ambiguous merging with other growing axons. Some axons split into two axons during growth, and at this point, the lower split axon was counted as the same axon, while the upper axon was counted as a start point for a new axon to analyze. The outputted angle had 0° defined as an orientation perfectly parallel with the underlying topographic pattern or parallel to a straight line arbitrarily drawn away from the DRG cell body on the 0 nm pitch control dishes. Axons that moved clockwise were a positive angle change from 0.1° to 180° and those moving counterclockwise were a negative angle change from -0.1° to −179.99°. This system was transformed to define 0° as both left and right on the *x*-axis emanating from the DRG body, with the *y* axis acting as +90° and −90° for up and down, respectively (see Appendix A for examples of live axon movement). Each axon had the growth angle measured for each frame. Those angles were averaged to determine the average angle of growth for each axon. This was measured from 6 DRG with a total of *n* = 95 axons on the 1400 nm pitch surfaces, and 5 DRG with a total of *n* = 87 axons on the flat, 0 nm pitch control surfaces.

### 2.6. Fabrication of Half-Tube Structures

NOA81 was coated onto a flat sheet of Teflon, and a flat control or topographically patterned PDMS stamp was placed onto the NOA81, topography-side down. The NOA81 was UV crosslinked for 400 s. The PDMS stamp was peeled off as previously described, and the NOA81 was peeled off from the Teflon film. The resulting NOA81 substrate was somewhat flexible. The NOA81 substrates were delicately placed into circular glass vials to consistently shape them into half-tubes, with the stamped topography on their inner walls. Vials with the NOA81 half-tubes within them were UV crosslinked for 6000 s. The half-tubes were carefully removed from the glass vials. The final products were rigid, half-tube structures with 1400 nm pitch, 4000 nm pitch or 0 nm pitch control surface topography on their 5 mm inner wall diameters. The half-tubes were glued onto 60 mm tissue culture plates with a small amount of NOA81 cured with UV for 5 min. Surface topography was verified using AFM in tapping mode.

### 2.7. Imaging DRGs on Half-Tube Structures

The half-tube structures were too thick for inverted fluorescence or confocal microscopes to obtain a clear image of the DRGs; thus, an upright Leica DM5500B fluorescence microscope (Leica Microsystems, Wetzlar, Germany) outfitted with a 1.4-megapixel monochrome CCD camera and a 10× objective was chosen for imaging. The DRGs were stained for β-tubulin with AlexaFluor-488 (Life technologies, Carlsbad, CA, USA) and imaged under 480-nm light with a GFP filter cube. Z-stack images were obtained using the Leica Application Suite AF (LAS AF) software (Leica Microsystems, Inc, Deefield, IL, USA, version 4.4), with a Z-interval of 2.766 µm. The tubes were aligned so that the long axis of the half-tubes pointed vertically. The field of view was manually moved with a precision mechanical stage and images were taken with an overlap of 5–10% with one another. The DRG body and at least one field of view left and right of the DRG body were captured, and the stage was moved in x and y from those points to capture the entire region of axon growth. The fluorescence Z-stack images of the DRGs on the half-tubes were deconvolved with theoretical point spread functions derived from the PSF Generator [42] and used with the DeconvolutionLab ImageJ plugin (Swiss Federal Institute of Technology Lausanne, Vaud, Switzerland, version 2.1.2) [43]. The Richardson–Lucy iterative deconvolution method was employed, and the resultant image stacks were then stitched together using the MosiacJ plugin (Swiss Federal Institute of Technology Lausanne, Vaud, Switzerland, version 2.0.0) [44] to create single large images for axon length quantification.

### 2.8. Statistical Analysis

Statistical analyses were performed with GraphPad Prism statistical software (GraphPad, San Diego, CA, USA, version 9). An unpaired *t*-test with Welch’s correction was used to evaluate the differences in mean axon velocity and mean axon growth angularity. The other experiments were assessed for statistical significance using one-way ANOVA with Tukey posthoc multiple comparisons tests. Significance was denoted at *p* < 0.05 *, or < 0.01 ** or < 0.001 ***.

## 3. Results

### 3.1. 1400 nm and 4000 nm Pitch Substrates Induce Directional Axon Regeneration from DRG Explants

Adult NIH-Swiss mouse-derived DRG were explanted on plates with individually patterned, repetitive ridge and groove pitches (pitch = ridge + groove) that were fabricated with soft lithography techniques (Figure 1A). The substrate patterns were all verified by atomic force microscopy (Figure 1B–E).

Chemically identical 0 nm pitch (flat) plates were used as the control. After six days, the explants were fixed, immunofluorescence labeled with anti-β-tubulin and anti-peripherin antibodies and imaged on a confocal microscope. The pattern of axonal growth varied with the topographic pitch sizes (Figure 2). DRG plated onto 0 nm pitch substrates always displayed radial axon sprouting from the DRG body with no preferred direction of growth (Figure 2A). On the 400 nm pitch surfaces, axon regeneration overwhelmingly displayed radial growth with an occasional axon that was parallel to the ridges and grooves (Figure 2B). A transition was observed with DRG on 800 nm substrates as they displayed a mixture of radial axon regeneration, but also had axons that were clearly following the direction of the underlying topographic pattern (Figure 2C). DRG plated onto 1400 nm and 4000 nm pitch substrates had axons that were nearly all parallel to the ridges and grooves (Figure 2D,E).

### 3.2. 1400 nm and 4000 nm Pitch Sizes Influence Axon Length and Direction

We quantified the day six confocal images of mouse DRG axon regeneration on the topographic surfaces and 0 nm pitch control using several metrics. At both 1000 µm and 1500 µm distances from the center of the DRG bodies (Appendix A), the 1400 nm and 4000 nm pitch surfaces caused nearly 100% of the axons to be parallel to the ridge and groove topography (Figure 3A,B). Our quantification method shows that axons grown on 800 nm pitch topography trend toward directional growth parallel to the underlying ridges and grooves, which was evident in the confocal images, but that directional growth was not statistically significant compared to DRG on 0 nm pitch, flat control plates.

Post peripheral nerve injury, surgical interventions may be necessary to induce long axon regeneration to restore sensory or motor function at a distal site. With that in mind, we quantified axonal length on the various patterned surfaces as the average of the three longest axons for each explant (Figure 3C). The 1400 nm pitch surface featured axons significantly longer compared to the 0 nm pitch surface and 400 nm pitch (*p* < 0.01). Average maximal axonal growth measures for all cases were as follows: 0 nm pitch (flat) = 2934 µm (*n* = 12 DRG), 400 nm pitch = 3066 µm (*n* = 19), 800 nm pitch = 3304 µm (*n* = 14), 1400 nm pitch = 4246 µm (*n* = 11) and 4000 nm pitch = 3502 µm (*n* = 12).

### 3.3. 1400 nm and 4000 nm Pitch Sizes Resulted in Greater Distal Axonal Density Compared to Control and 400 nm Surfaces

Besides maximal axonal length, distal axonal density may be necessary to fully restore function after peripheral nerve damage. As such, distal axonal density in our experiments was quantified as the number of axons at the 1500 µm boundary in the left and right counting quadrants parallel to the topography or to an arbitrary axis for the 0 nm pitch control (Figure 3D). Both the 1400 nm (*p* < 0.05) and 4000 nm (*p* < 0.01) pitch surfaces had a significantly higher axon density 1500 µm from the DRG body compared to the other topographies or 0 nm pitch control. However, no statistical difference was observed between axon density on 1400 nm versus 4000 nm pitched topography.

### 3.4. Axons Regenerating on 1400 nm Pitch Surface Maintained Parallel Directionality Irrespective of Neurotrophic Stimulus

In Figure 2 and Figure 3, the mouse DRG explants were grown ex vivo in media containing NGF for 6 days. We wanted to test the hypothesis that underlying topographical patterns acted as the dominant directional growth cues for the regenerating axons. If soluble neurotrophins, evenly diffused throughout the media were the dominant directional growth cue, then we would expect a more even, radial pattern of axon growth. When DRG explants were grown on either the flat 0 nm pitch or the 1400 nm topographical pitch surfaces with one neurotrophin (either NGF, GDNF, BDNF or NT-3) in the media, nearly 100% of the distal axons were parallel to the 1400 nm pitched topography 6 days after plating, regardless of the neurotrophin used (Figure 4). Additionally, DRG grown on 1400 nm plates with NGF had average axon lengths that were statistically longer than DRG with any of the other three neurotrophins tested on flat or 1400 nm pitch surfaces (data not shown).

### 3.5. Axon Growth Angles Are Directed by Sub-Micron Sized Surface Features

The data from Figure 2, Figure 3 and Figure 4 featured DRG that were stained 6 days post-plating. These data were obtained after fixing the plate, which did not provide axon growth dynamics in real-time. To gain a better understanding of how axons behaved as they were growing on the topographical pitch surfaces, live-cell imaging was performed with phase contrast to measure axon growth angles and speeds (Appendix A). This would determine whether the main driving force for the enhanced axon growth length on topography was an increased growth speed, efficiency of growth directionality or both. Axon growth speed was found to have no significant difference between DRG growing on the 1400 nm surface (41.45 µm/h) compared to the flat 0 nm pitch control surfaces (38.63 µm/h) (*p* = 0.1601, Figure 5A). Histograms of the tracked axon populations show a similar distribution of axon speeds on the 0 nm pitch and 1400 nm pitched surfaces (Figure 5B,C).

The tips of individual axon growth cones were manually tracked in Image J, frame-by-frame, and the angles of each of those steps were measured and averaged for each axon. If an axon grew perfectly parallel to the underlying ridges and grooves, or to an arbitrary axis on the 0 nm pitch controls, the axon was assigned a growth angle of 0°. Deviations to the topographic axis were measured as positive and negative angles of growth (Figure 6A). The average growth angles were significantly different between the 1400 nm topographic pitch and 0 nm pitch (flat) surface types (*p* < 0.05) (Figure 6B). Histograms revealed that angles of axon growth on the 0 nm pitch control were, as expected, far more varied (Figure 6C) compared to the 1400 nm topography (Figure 6D).

### 3.6. Directional DRG Explant Growth Was Similar on Topographically Patterned 3-Dimensional Tubes

Ultimately, conferring directionality to the axonal growth is aimed at intelligently designing 3-D scaffolds with the ability to guide axon regeneration for clinical purposes. An initial step towards that goal was to test how fibers grow in a hemi-3-D model. We then wanted to know if the results of our findings could translate to truly 3-D materials. As such, we fabricated half-tube structures (Figure 7A,B) with either topographic ridge and groove inner walls or smooth, 0 nm pitch (flat) control inner walls (Figure 7C,D). DRG were seeded onto the half-tube structures, just as before, and were grown for 6 days, fixed and viewed with an upright widefield fluorescence microscope to create stitched images. We fabricated half-tube structures instead of complete tubes to allow for easier microscopic visualization of the DRG axons. Once again, we observed strong axon alignment to the underlying 4000 nm and 1400 nm topographically pitched half-tubes (Figure 7E,F). In contrast, the 0 nm pitch (flat) control half-tubes induced radial patterns of axon growth (Figure 7G). Furthermore, we found that axon growth length was significantly increased on the 4000 nm pitch half-tube compared to both the DRG on the 0 nm pitch (flat) control half-tubes (*p* < 0.001) and 1400 nm pitch half-tubes (*p* < 0.01). The DRG axons grown on the 1400 nm half-tubes were significantly longer than the 0 nm pitch (flat) control (*p* < 0.05) (Figure 7H). Taken together, these results demonstrate that regenerating DRG axons will align with 1400 nm and 4000 nm pitched ridge and grove patterns and as a result, axons will grow further from the DRG body, even within 3-dimensional half-tube scaffold structures.

## 4. Discussion

Peripheral nerve injury is a potentially devastating sequela seen in 2–3% of trauma patients [45]. Clinical outcomes after the most severe of these injuries (neurotmesis), with loss of nerve material, range from full functional recovery to persistent pain, to lack of ability to perform activities of daily living. The gold standard in gap repair remains the autograft, which carries with it the cost of significantly increased surgical time and donor-site morbidity [46]. Often, surgeons rely on several options for off-the-shelf reconstruction in the operating room, such as conduits or wraps and acellularized allografts. While these biomaterials do work with various levels of efficacy, they also have limitations. Augmenting these materials with appropriate biophysical cues could potentially improve outcomes.

Previous studies have demonstrated that biophysical cues incorporated into various substrates can influence neuron regeneration [22,33,34,47]. However, the style, shape and feature sizes of the cues on the substrate are critical considerations. Neurons can survive, adhere, migrate and orient their axons to different biophysical surface cues [48]. Nevertheless, these studies have mostly utilized cultured cell lines and/or topographic channels much larger in scale than biomimetic extracellular matrix geometries. In vitro models, namely, single-cell systems, can, unfortunately, be too simplistic to correlate to physiologically relevant events. On the other hand, in vivo models are a challenge to control due to immune responses, glial scarring after injury and the intricate complexity of the surrounding extracellular matrix. The DRG organotypic culture used for this study offers an attractive multicellular option to model the human pathophysiology of fiber regeneration. The combination of the explant with modular topographic patterns of different sizes creates an ex vivo system that is easy to manipulate with extracellular protein coatings, and the addition of neurotrophins and other soluble molecules. The ex vivo explant model of mouse dorsal root ganglia used for this proposal mimics the real cell–cell interaction and biochemical cues surrounding growing axons [49]. The harvesting of the DRG in vivo and explantation in vitro reproduces the damage condition axonotmesis, in which axons are fully severed and the neuronal cell body is no longer connected with the innervated target.

In this study, we used soft lithography techniques to create substrates with anisotropic ridge and groove patterns. The ridges and grooves were in a 1:1 ratio with pitch sizes that ranged from 400 nm to 4000 nm. When whole mouse DRG explants were plated onto 1400 nm and 4000 nm pitch sizes, we observed axon sprouting and growth that was nearly 100% aligned with the underlying surface topography. Importantly, the size of the ridge and groove features determined the amount of axon alignment. On a 400 nm pitch, the axon growth was similar to the radial growth patterns seen on chemically identical flat, 0 nm pitch control surfaces. DRG on the 800 nm pitch surfaces displayed some axonal growth parallel to the ridge and groove topography based on our confocal images, but it was not statistically different from the 400 nm pitch or 0 nm pitch surfaces. This suggests that regenerating axons find those smaller pitch sizes, which have individual ridge and groove widths of either 200 nm or 400 nm, somewhat indistinguishable from smooth surfaces. This behavior may be explained by the size of nerve fibers in the mouse model.

Nerves in mice (lingual, intermediate and facial) have been measured by stereological measurements using electron microscopy. The diameter of unmyelinated fibers ranged from 0.2 to 1.2 µm [50,51,52]. Myelinated fibers are bigger in size with 0.4–3 µm diameter of axons. If fibers utilize ridges as a track to guide axonal growth, the 1400 nm and 4000 nm pitch surfaces, which feature 700 nm and 2000 nm ridge widths and are spaced apart by 700 nm and 2000 nm grooves, may provide an ideal size ratio of biophysical information to serve as a track for directional axon growth. The topographic substrates with ridges of 200 nm widths (0.2 µm) used for our 400 nm pitch topography surfaces did not induce directional growth, maybe because the large fibers “surfed” over the grooves. The 800 nm pitch featuring a ridge of 400 nm (0.4 µm) may appeal to some smaller unmyelinated fibers but still may not be wide enough for the larger myelinated fibers.

Testing pitch sizes larger than 4000 nm was beyond the scope of this study. However, other published work using similar ridge scaffold sheets that were tens of microns wide, but rolled up into looped tubes, observed rat DRG axon growth that was both parallel and oblique to the topographic patterns [53]. Decreases in the sizes of the grooves within the looped sheets resulted in more parallel axon growth. Interestingly, chick DRG have been plated onto polymeric spun fibers with diameters of 10 µm–100 µm. Those dimensions resulted in axon growth that was parallel to the aligned substrate fiber mats, but the parallel axon orientation was most consistent on the relatively smaller, 10 µm diameter fibers [54]. Additionally, other work has shown that ridge and groove features 15–100× larger than ours (30 µm grooves and 200 µm ridges) can stimulate rat axons to grow perpendicular to topographic grooves [34]. Besides our DRG ex vivo cultures, another study plated dissociated cortical embryonic rat neurons onto substrates with sine wave-like periodic patterns with 1–6 µm pitch sizes, which is slightly different to our square wave-like patterns. The dissociated neurons produced axons that were tightly aligned with 4 µm and 5 µm pitch sizes, and that alignment was abolished by taxol and blebbistatin treatments that disrupted the cytoskeleton [31]. Taken together, this suggests that there may be an ideal “sweet spot” when designing biomaterials with ridge and groove feature sizes to optimally direct nerve regeneration by controlling cytoskeleton arrangements.

It was unclear if the longer axon growth on the 1400 nm pitch topography in our studies was due to the enhanced speed of axonal growth or efficient directional growth. Our live cell imaging results suggest that axon regeneration speeds do not change on the 1400 nm pitch surface compared to flat control surfaces. Instead, the data strongly suggest the driving force for the enhanced length of axonal outgrowth on the 1400 nm and 4000 nm pitch size was the efficiency of growth direction, ensuring axons minimize growth in undesired directions. We had similar directional axon regeneration results on our half-tube structures. The promise of these experiments is the eventual possibility of clinically useful, implantable conduit biomaterials with the appropriate surface topography fabricated into the inner walls. Such a material could be used to enhance peripheral nerve regeneration when surgically introduced into a traumatic injury site and have the advantage of altering and controlling cell behaviors via an intrinsic part of its structure, rather than a slow-release biochemical factor that may diffuse into the environment, losing its chemotaxis properties over a short period of time.

## 5. Conclusions

The DRG organotypic cultures plated onto well-defined, topographically patterned substrates used for this study offer an attractive multicellular option to model peripheral nerve pathophysiology. Our modular ex vivo system can be easily manipulated with extracellular protein coatings, soluble molecules such as neurotrophins and other activators or inhibitors and can be fabricated with a range of submicro- to micro-scale topographical patterns. With this highly modular culture system and reductionist experiments, we can further define the temporal and spatial changes and key signaling pathways that help direct axon regeneration after injury. Additionally, future studies could employ full-tube structures with interior surface topographies controlling cell behavior. This should both provide insights as to how axons navigate their surrounding extracellular matrix and aid in the design or discovery of implantable biomaterials that have appropriately sized shapes and anisotropic patterns to guide damaged neurons back to the correct tissue targets.

## Figures and Tables

**Figure 1 biomolecules-12-01059-f001:**
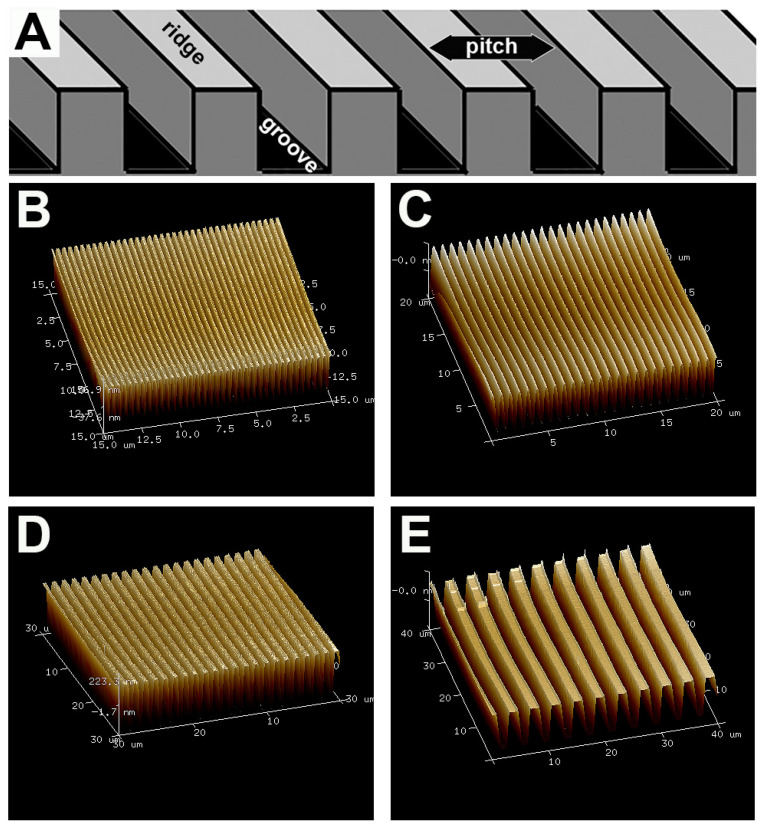
Diagram and AFM imaging of substrate surface topography. (**A**) A schematic of the topographic surfaces used in this study. Pitch size is equal to the width of one ridge plus one groove in a 1:1 ratio. Atomic force microscopy was used to inspect the integrity of the surface topography. AFM images of the different pitched substrates used in this study: (**B**) 400 nm substrate, (**C**) 800 nm substrate, (**D**) 1400 nm substrate and (**E**) 4000 nm substrate. Depth of the grooves was 300 nm.

**Figure 2 biomolecules-12-01059-f002:**
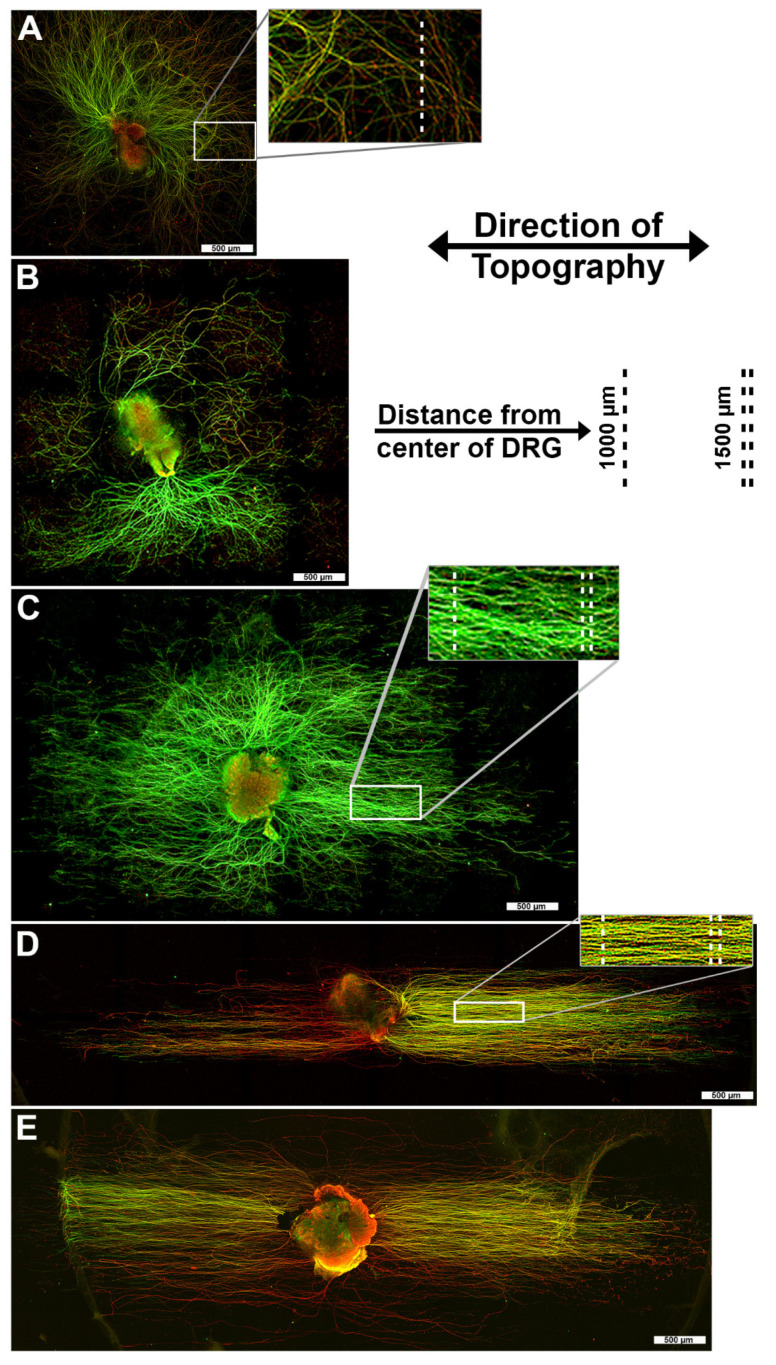
DRG explants cultured on surfaces with topography of varied pitch sizes. Dorsal root ganglion cultured for 6 days ex vivo on ridge and groove topographies of varying pitch sizes exhibit different growth patterns. All topographic ridge and groove features are oriented horizontally, except for the flat, 0 nm pitch control that has no orientation. DRG cultured on (**A**) 0 nm pitch, (**B**) 400 nm pitch and (**C**) 800 nm pitch topography. These feature sizes produced a radial pattern of axon regeneration with the 800 nm pitch surface inducing a limited amount of visible, directional growth. DRG cultured on (**D**) 1400 nm pitch topography had axons that extended further from their origin compared to a flat surface. The DRG exhibited a linear and directional pattern of axon regeneration parallel to the underlying 1400 nm pitch surface topography. DRG cultured on (**E**) 4000 nm pitch topography also displayed axonal growth parallel to the surface pattern. DRG were stained with primary antibodies recognizing tubulin (green) and peripherin (red) and imaged with laser scanning confocal microscopy. Inset panels show detailed axon orientation at both 1000 µm and 1500 µm from the center of the DRG body. A-E scale bars = 500 µm.

**Figure 3 biomolecules-12-01059-f003:**
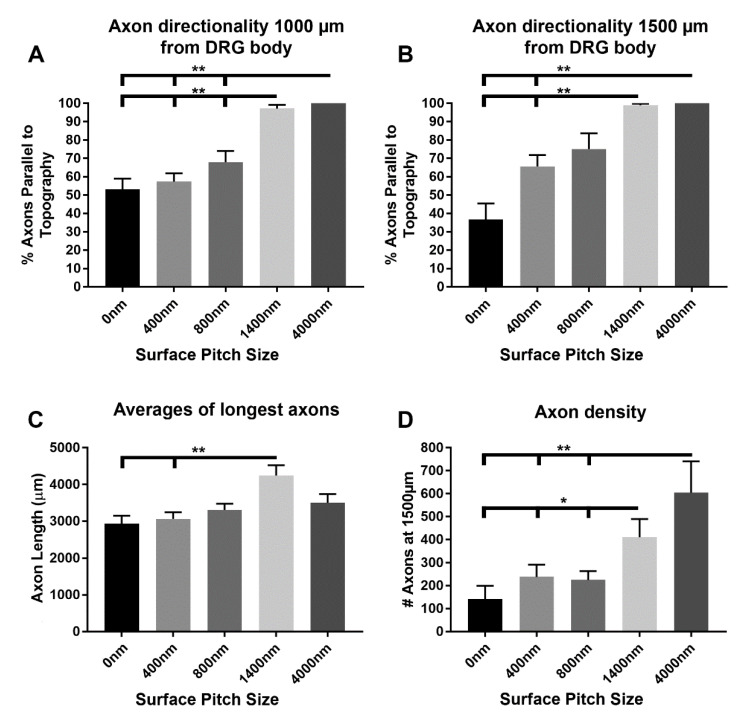
Quantitative measurements of DRG axon directionality, length and density 6 days post-plating on the topographic surfaces with various pitch sizes. The directionality of axon growth in relation to the orientation of the ridge and groove patterns of the substrates was measured (**A**) 1000 µm from the DRG body and (**B**) 1500 µm from the DRG body. Nearly 100% of the regenerated axons were parallel to the 1400 nm and 4000 nm pitch topography at both 1000 and 1500 µm from the DRG body, which was significantly different from the other surfaces (*p* < 0.01) with the exception of 800 nm pitch surface at 1500 µm. (**C**) Maximal axonal growth on topographic surfaces with various pitch sizes was measured from the DRG center point in a straight-line vector to an axon terminus. The three longest vectors were quantified and averaged together for each explant. The axons on the 1400 nm pitch surface had significantly increased growth compared to the 0 nm pitch and 400 nm pitch surfaces (*p* < 0.01). (**D**) Axonal density on topographic surfaces was measured as the number of axons crossing a boundary drawn 1500 µm from the DRG center. The 1400 nm and 4000 nm pitched topographic surfaces resulted in greater axon density than the 0 nm, 400 nm and 800 nm pitched topography (* = *p* < 0.05 and ** = *p* < 0.01, respectively). Statistical differences were determined by ANOVA with Tukey posthoc analysis.

**Figure 4 biomolecules-12-01059-f004:**
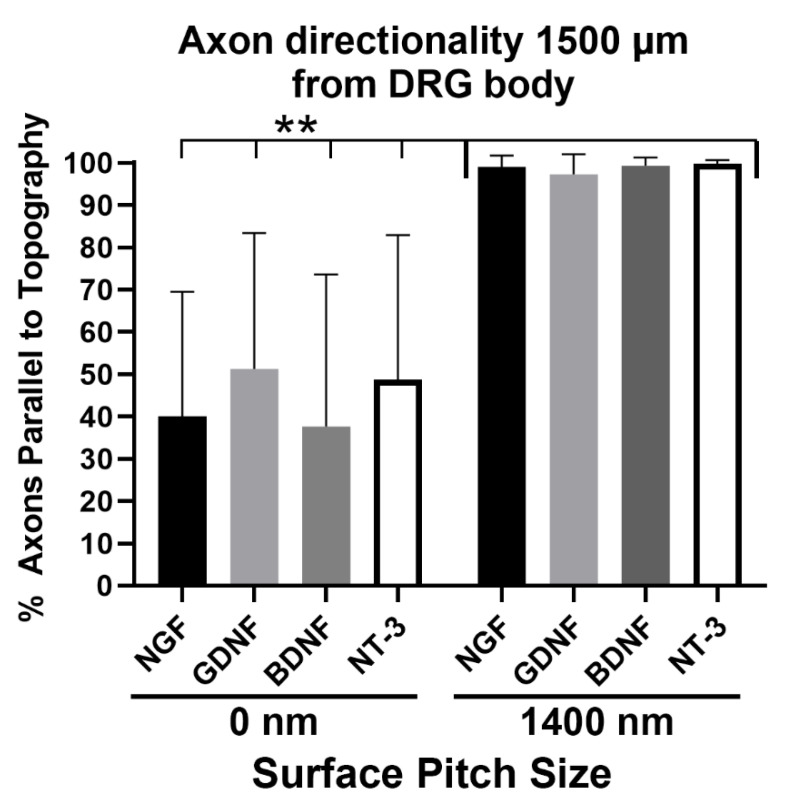
Axon directionality on 1400 nm topographical pitch surface or 0 nm pitch flat surface is not altered by diffuse soluble factors. DRG were plated onto either 0 nm pitch control surfaces or 1400 nm pitch surfaces with a basal media enriched with either NGF, GDNF, BDNF or NT-3. Six days after plating, DRG were fixed, stained and imaged using confocal microscopy. The directionality of axons 1500 µm from the DRG body was quantified. DRG cultured on the 1400 nm pitch topography had 85–100% of their axons parallel to the underlying surface topography, independent of the soluble growth factor that was added to the media. The DRG cultured on the 0 nm pitch control substrates had individual axons that ranged from 0–100% parallel to an arbitrary, consistent axis. Each 1400 nm pitch category was significantly different (** = *p* < 0.01) from each control 0 nm pitch (flat) category based on an ANOVA with Tukey posthoc analysis. The number of DRG (*n*) for each condition: NGF 0 nm pitch *n* = 11, GDNF 0 nm pitch *n* = 10, BDNF 0 nm pitch *n* = 9, NT-3 0 nm pitch *n* = 13, NGF 1400 nm pitch *n* = 11, GDNF 1400 nm pitch *n* = 13, BDNF 1400 nm pitch *n* = 9 and NT-3 1400 nm pitch *n* = 10.

**Figure 5 biomolecules-12-01059-f005:**
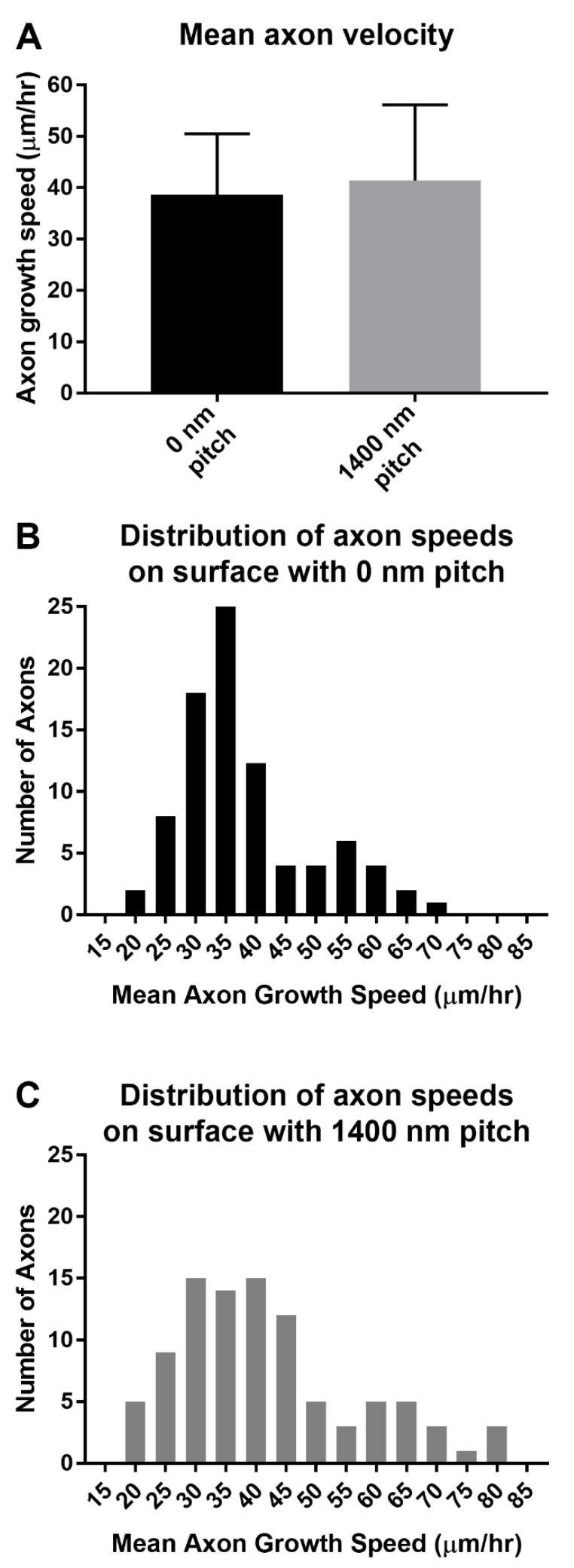
Speed of DRG axon growth on flat, 0nm pitched and 1400 nm pitched topographic surfaces was similar. The velocities of regenerating axons were measured using time-lapse phase contrast microscopy between 72–96 h after plating DRG on 1400 nm pitch topographic plates (95 axons from 6 DRG) and flat, 0 nm pitch controls (87 axons from 5 DRG). (**A**) The mean axon growth velocities were not statistically different (*p* = 0.1601). Histogram distributions of axon regeneration speeds on (**B**) 0 nm pitch controls versus (**C**) 1400 nm pitched surfaces were also similar.

**Figure 6 biomolecules-12-01059-f006:**
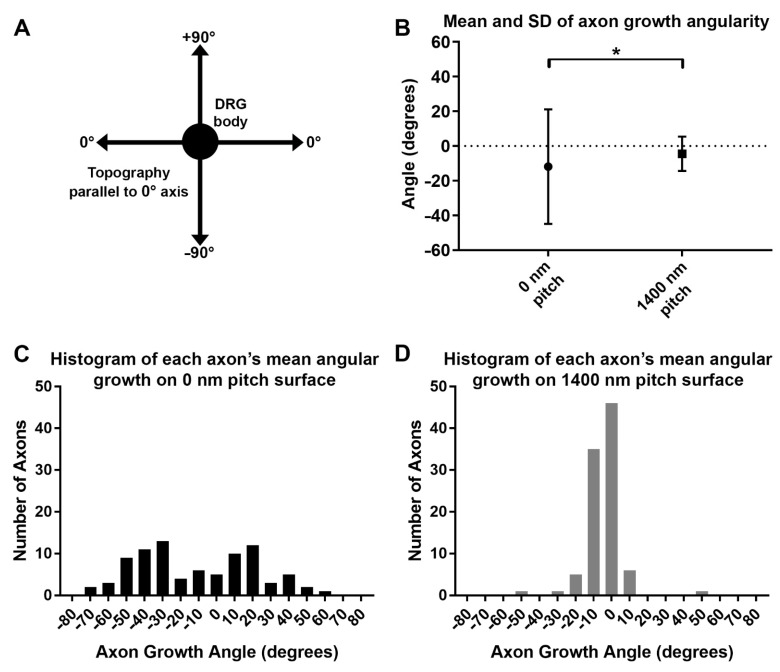
The angularity of regenerating DRG axon growth was directed by topographic substrate patterns. (**A**) A schematic demonstrating how the angles of each step of axon growth were measured with respect to the DRG body. (**B**) The means and standard deviations of axon growth angularity measured on 0 nm pitch (flat) control surfaces (87 axons from 5 DRG) versus 1400 nm pitch surfaces (95 axons from 6 DRG) (*t*-test, * = *p* < 0.05) show that axons regenerating on 1400 nm pitched surfaces tend to deviate less than ±10° from the direction of the topographic grooves while axons on the 0 nm pitch control have an angular spread nearly three times that. (**C**) A histogram of the angular growth of axons on 0 nm pitch surfaces has a highly variable distribution compared to the (**D**) histogram distribution seen from axons growing on 1400 nm pitched surfaces.

**Figure 7 biomolecules-12-01059-f007:**
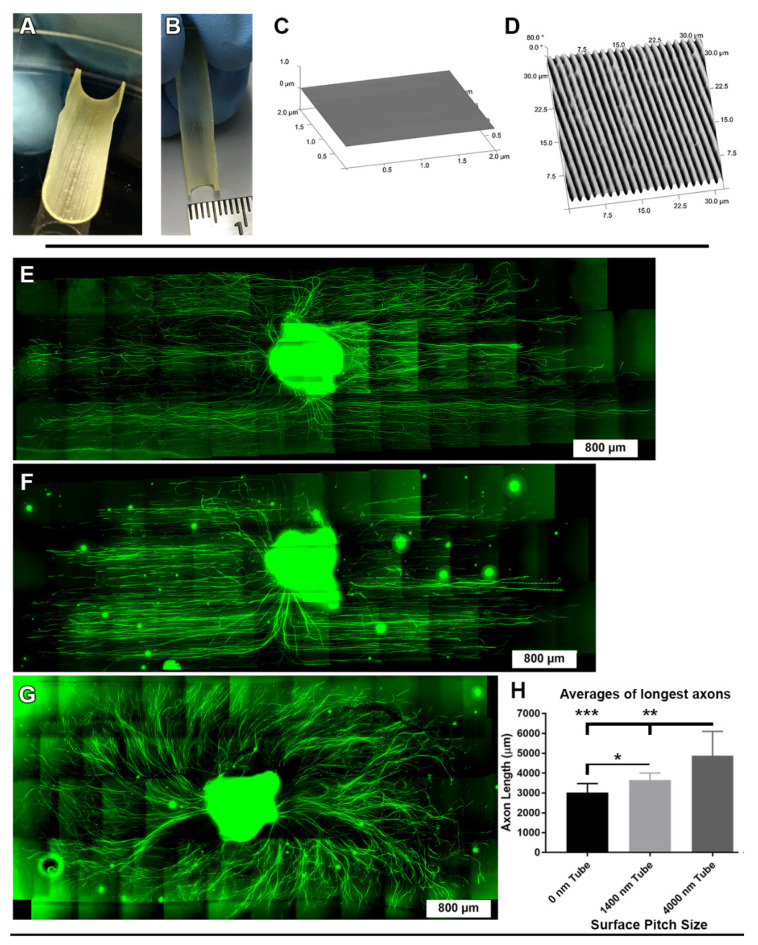
DRG axons regenerate parallel to grooved topography on 3-D tube structures. (**A**) Semi-circular half-tubes with topographically patterned interior surfaces were fabricated to have (**B**) a 0.5 cm inner diameter. AFM was used to verify surfaces of (**C**) 0 nm pitch (flat) (**D**) 1400 nm pitch or 4000 nm pitch (not shown). DRG were plated onto the center of the tubes and allowed to grow, covered in media, for 6 days. After 6 days the DRG were fixed, immunostained for tubulin (green), and imaged with stitching using an upright fluorescence microscope. DRG that were cultured on (**E**) 4000 nm pitch or (**F**) 1400 nm pitch semi-circular tubes displayed axonal growth that was primarily parallel to the underlying surface topography. The DRG cultured on (**G**) control 0 nm pitch half-tubes had axons that radiated away from the DRG body in all orientations. (**H**) The DRG cultured on the 4000 nm pitch half-tubes produced regenerative axons that were significantly longer than those from the DRG on the 1400 nm pitch (** *p* < 0.01) or 0 nm pitch control half-tubes (*** *p* < 0.001). The DRG axons cultured on the 1400 nm pitch half-tubes were significantly longer than the DRG on the 0 nm pitch half-tubes (* *p* < 0.05). (**E**–**G**) scale bars = 800 µm.

## Data Availability

Not applicable.

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
