# Peer review of "Submicron Topographically Patterned 3D Substrates Enhance Directional Axon Outgrowth of Dorsal Root Ganglia Cultured Ex Vivo"

_biomolecules, 2022, doi:10.3390/biom12081059_

Round 1

Reviewer 1 Report

In this manuscript the authors study the outgrowth of axons from adult DRG explants on micropatterned substrates. They find that substrates with parallel grooves and ridges of a width of 1.4 and 4 um induce parallel axonal growth, while similar substrates with a width of 1 um and less do this not or much less. Due to the aligned straight outgrowth pattern the axons achieve longer distances compared to outgrowth on an unpatterned surface.

The manuscript is well organized and apart from some minor language problems is easy to read and understand. The results are well documented and presented and are interesting for researchers in the field of peripheral nerve regeneration. This reviewer has only a few points which the authors should address in order to improve the manuscript.

1) Discussion of the relevant literature. The authors suggest the few studies with a similar approach exist, quote “Nevertheless, these studies have mostly utilized cultured cell lines and/or topographic channels much larger in scale than biomimetic extracellular matrix geometries.” This is not case. In the recent publication by Sunnerberg et al. (2021), DOI: 10.1371/journal.pone.0257659, a very similar paradigm has been used with very similar findings. This paper should be mentioned and the relevant findings should be discussed in the manuscript.

Similarly, also the less recent papers by Omidinia-Anarkoli et al. 2020, DOI:10.1016/j.actbio.2020.07.014 and Tuft et al. 2013, DOI:10.1016/j.biomaterials.2012.09.053 need to be mentioned and discussed by the authors.

2) Discussion of the role of Matrigel. As the patterned surfaces were coated with the 3D component Matrigel it would be useful to repeat the AFM scanning microscopy after coating with Matrigel. It can be expected that the geometry of the ridges and grooves is considerably altered by this coating. Since the growth cones will only be exposed to the coated surfaces, the real topography of the coated surface is of considerable interest.

Author Response

In this manuscript the authors study the outgrowth of axons from adult DRG explants on micropatterned substrates. They find that substrates with parallel grooves and ridges of a width of 1.4 and 4 um induce parallel axonal growth, while similar substrates with a width of 1 um and less do this not or much less. Due to the aligned straight outgrowth pattern the axons achieve longer distances compared to outgrowth on an unpatterned surface.

The manuscript is well organized and apart from some minor language problems is easy to read and understand. The results are well documented and presented and are interesting for researchers in the field of peripheral nerve regeneration. This reviewer has only a few points which the authors should address in order to improve the manuscript.

1) Discussion of the relevant literature. The authors suggest the few studies with a similar approach exist, quote “Nevertheless, these studies have mostly utilized cultured cell lines and/or topographic channels much larger in scale than biomimetic extracellular matrix geometries.” This is not case. In the recent publication by Sunnerberg et al. (2021), DOI: 10.1371/journal.pone.0257659, a very similar paradigm has been used with very similar findings. This paper should be mentioned and the relevant findings should be discussed in the manuscript.

Similarly, also the less recent papers by Omidinia-Anarkoli et al. 2020, DOI:10.1016/j.actbio.2020.07.014 and Tuft et al. 2013, DOI:10.1016/j.biomaterials.2012.09.053 need to be mentioned and discussed by the authors.

Response to reviewer 1:   We have modified our intro with the addition of suggested references. We also added to our discussion by citing the 2021 article by Sunnerberg et al and 2020 article by Omidinia-Anarkoli, along with presenting a summary of their findings in the context of ours.  We thank the reviewer for pointing out these other studies to compare to ours.

2) Discussion of the role of Matrigel. As the patterned surfaces were coated with the 3D component Matrigel it would be useful to repeat the AFM scanning microscopy after coating with Matrigel. It can be expected that the geometry of the ridges and grooves is considerably altered by this coating. Since the growth cones will only be exposed to the coated surfaces, the real topography of the coated surface is of considerable interest.

Response to reviewer 1: Although the Matrigel may slightly modify the surface topography, we believe the effects are likely to be minor and it would be a challenge to reliably image those subtle changes in the surface topography with an AFM.  The Matrigel was diluted to 0.5X in media, and only 10 µl was used as a “glue” to help the DRG body adhere to the surface.  Therefore, the diluted Matrigel was located primarily around the DRG body and was one of the reasons we focused all of our quantitative axon growth metrics beyond 200 µm from the DRG center. All of the surfaces, including the flat controls, received the same 10 µl of 0.5X Matrigel, so there was consistency in all of our platings. Based on phase contrast images that can be seen in the supplemental movies, there is no evidence of submicron-sized surface irregularities caused by the thin Matrigel coating.  Undoubtedly, the Matrigel caused some changes to the surfaces topography, especially near DRG body, but beyond a 200 µm radius the changes were likely to be nanometer-sized and our conclusion is that ridge and groove pitch sizes between 1400 nm and 4000 nm induce the directional axon growth.  Alterations by a few nanometers are unlikely to cause different results in that range.  While we do plan to eventually image the Matrigel coating with precise AFM techniques, we do not believe those potentially small surface changes are significantly driving the axon growth behaviors we observed.  We have now briefly addressed this question in the manuscript in the methods section that describes DRG plating.

Reviewer 2 Report

Paper entitled Submicron topographically patterned 3D substrates enhance directional axon outgrowth of dorsal root ganglia cultured ex vivo” is well-written and the contents seems very interested. Authors shown that dorsal root ganglia (DRG) organotypic cultures can grow their axons ex vivo in various manner, depending on surface shape and environmental conditions.  They selected 1400 nm pitch surface as the best in terms of axons length, and also significant better in axons density comparing to flat surface. They also tested various media enrichments (NGF, GDNF, BDNF and NT-3), compering 1400 nm pitch surface vs flat surface, showing improvement of the directionally of axons 1500 µm from the DRG body in 1400 nm pitch surface regardless of media enrichment.  All of the presented results are very interesting, and I am looking forward to read future in vivo studies using presented in this paper approaches.  

However, suggests adding a few editorial amendments to the manuscript:

1.      Figure 3, 5, 6,  – information about the “n” and used statistical test is missing

2.      Figure 4 – information about the “n” is missing

3.      Figure 2 and 7 – insets presenting axons morphology and density at the distance 1000 and 1500 µm from DRG body will illustrate better presented results

Present paper is valuable source of information, with correct designed experiments and clearly showed results and I highly recommend it for publication after minor corrections. 

Author Response

Suggested adding a few editorial amendments to the manuscript:

  1. Figure 3, 5, 6,  – information about the “n” and used statistical test is missing.

Response to reviewer 2: The n numbers and statistical tests were already included in the Methods and Results sections for Figure 3. The figure legend has been modified to include the statistical test details used in Figure 3.  The n numbers for Figure 5 were already present in the figure legend and there was no statistical difference between the groups, as stated in the legend.  The n numbers for Figure 6 are the same as Figure 5 and that has now been made more clear in the Figure 6 legend. The t-test is now also specifically mentioned in the legend.

  1. Figure 4 – information about the “n” is missing

Response to reviewer 2: We apologize for that oversight.  We have now included all the DRG n numbers in the Figure 4 legend.

  1. Figure 2 and 7 – insets presenting axons morphology and density at the distance 1000 and 1500 µm from DRG body will illustrate better presented results

Response to reviewer 2: Insets have been generated for Figure 2 confocal microscopy images of DRG on the flat, 800 nm and 1400 nm surfaces to demonstrate the axon growth orientation and density at 1000 µm and 1500 µm from the DRG body.  The Figure 2 legend has been modified to explain this in detail. However, we did not include insets for Figure 7 because those experiments required a manually stitched upright fluorescence microscope that was not as accurate as our inverted, automated confocal microscope.  We felt our longest axon measurements were indeed accurate using the upright fluorescence microscope, but we were unsure of the accuracy of axon density measurements on the challenging-to-image tube-like surfaces.  That is why we did not perform density measurements at 1000 µm and 1500µm distances with those samples, and thus why we did not include insets at 1000 µm and 1500µm.